# Targeted Therapy for Anaplastic Thyroid Carcinoma: Advances and Management

**DOI:** 10.3390/cancers15010179

**Published:** 2022-12-28

**Authors:** Jiaqian Yuan, Yong Guo

**Affiliations:** 1The First Clinical Medical College, Zhejiang Chinese Medical University, Hangzhou 310053, China; 2Department of Medical Oncology, The First Affiliated Hospital of Zhejiang Chinese Medical University, Hangzhou 310001, China

**Keywords:** anaplastic thyroid carcinoma (ATC), targeted therapy, BRAF/MEK inhibitors

## Abstract

**Simple Summary:**

The exceedingly aggressive and rare tumor, anaplastic thyroid carcinoma, is generally resistant to traditional antitumor therapies. In recent years, targeted therapy has become a hot research topic, giving patients with this malignant disease great hope. Therefore, it is necessary to review these studies and summarize the efficacy and adverse effects of various types of targeted drugs, not only to provide practical information for future basic research on related targeted drugs but also to help clinicians understand the research advances and management of these drugs in order to make the best clinical treatment recommendations for patients.

**Abstract:**

Anaplastic thyroid carcinoma (ATC) is a rare and highly fatal cancer with the worst prognosis of all thyroid carcinoma (TC) histological subtypes and no standard treatment. In recent years, the explosion of investigations on ATC-targeted agents has provided a new treatment strategy for this malignant condition, and a review of these studies is warranted. We conducted a comprehensive literature search for ATC-targeted drug studies and compiled a summary of their efficacy and adverse effects (AEs) to provide new insights. Multiple clinical trials have demonstrated the efficacy and safety of dabrafenib in combination with trametinib for the treatment of ATC, but vemurafenib and NTRK inhibitors showed limited clinical responses. We found that the previously valued therapeutic effect of lenvatinib may be unsatisfactory; combining tyrosine kinase (TK) inhibitors (TKIs) with other agents results in a higher rate of clinical benefit. In addition, specific medications, including RET inhibitors, mTOR inhibitors, CDK4/6 inhibitors, and Combretastatin A4-phosphate (CA4P), offer tremendous therapeutic potential. The AEs reported for all agents are relatively numerous but largely manageable clinically. More clinical trials are expected to further confirm the effectiveness and safety of these targeted drugs for ATC.

## 1. Introduction

Anaplastic thyroid carcinoma (ATC) is a rare malignant tumor comprising approximately 1–2% of thyroid carcinoma (TC) pathological types [1]. ATC is highly aggressive and has a dismal prognosis, with a median survival of about 5 months and 1-year overall survival (OS) of 20% [2], which is more prevalent in women. Patients with this disease may have a history of long-term goiter rapidly enlarging in a short period, presenting with neck pain, a hard and fixed mass with poorly defined borders, and a propensity to quickly develop symptoms such as dyspnea, hoarseness, and other symptoms arising from compression of the trachea and recurrent laryngeal nerve. The majority of patients with ATC have infiltration and distant metastases at the time of initial diagnosis, missing the optimal window for surgery.

The signaling pathways in ATC mainly involve MAPK, PI3K/AKT/mTOR, and JAK-STAT, which include important targets such as RET, EGRF, and KIT, as shown in Figure 1. Mutation of a target deranges downstream signaling, leading to dysregulation of cell growth and apoptosis and, ultimately, cancer development. Targeted therapy has become a highly valued treatment, which inhibits the abnormal proliferation of malignant cells by blocking specific signaling pathways.

The overall tumor mutation burden of ATC is greater than other types of TC, with BRAF and RAS oncogenes, which are components of the RAS-to-ERK signaling pathway, being the main drivers [3]. It has also been shown [4] that ATC or poorly differentiated thyroid carcinoma (PDTC) arises from the evolution of well-differentiated thyroid tumors, primarily through the gradual accumulation of additional genetic abnormalities, that the most typical genetic alteration leading to the development of this phenomenon is the loss of TP53 tumor suppressor, and that TP53 mutations also affect the efficacy of chemotherapy and radioiodine therapy [5]. However, P53 inactivation, considered a hallmark of advanced thyroid cancer, is extremely rare in differentiated thyroid carcinoma (DTC). In addition, mutations in the TERT promoter, PI3K/AKT, and mTOR pathway effectors are also common in ATC. Clinically, ATC is exceptionally aggressive, and unlike DTC, it is resistant to conventional tumor therapies such as radiotherapy and chemotherapy, making treating such a malignant condition more challenging. With the rise of precision therapy, targeted therapy for ATC has also developed dramatically in recent years. A retrospective cohort study [6] evaluated the survival of patients with ATC over 20 years from 2000 to 2019, which showed that targeted therapy significantly improved the OS of patients with ATC compared to those who did not receive targeted therapy (*p* < 0.001). American Thyroid Association (ATA) guidelines for ATC in 2021 recommend genetic testing at the first sign of ATC suspicion [2]. We focused on recent research, summarized in vitro and in vivo studies of several critical targeted agents for ATC, and analyzed the adverse events (AEs) caused by related targeted drugs in textual or tabular (if there are two or more clinical trials of a drug specific to ATC patients) form to provide information for the basic research of ATC-targeted drugs and assist clinicians in recognizing the research advances and management of these related drugs.

## 2. Materials and Methods

We searched Pubmed, Embase, Web of Science, and Medline from inception to 30 September 2022. The following search terms were used: ‘anaplastic thyroid carcinoma’, ‘anaplastic thyroid cancer’, ‘ATC’, ‘undifferentiated thyroid cancer’, ‘targeted drug’, ‘targeted medicine’, and ‘targeted agent’. The relevant journals, bibliographies, and reviews were manually searched for additional articles. Inclusion criteria: (1) clinical trials of targeted drugs for ATC, (2) in vitro and in vivo trials of targeted drugs for ATC, and (3) meaningful cases of targeted drug therapy for ATC. Exclusion criteria: (1) duplicate literature, (2) repetitive research, (3) trials of other diseases, (4) irrelevant reviews, (5) research at the molecular or genetic level, and (6) clinical trials of thyroid cancer excluding ATC.

The search identified 735 references in total. After screening, 663 studies were discarded, leaving 72 studies finally [7,8,9,10,11,12,13,14,15,16,17,18,19,20,21,22,23,24,25,26,27,28,29,30,31,32,33,34,35,36,37,38,39,40,41,42,43,44,45,46,47,48,49,50,51,52,53,54,55,56,57,58,59,60,61,62,63,64,65,66,67,68,69,70,71,72,73,74,75,76,77,78], detailed in Figure 2. These original papers were read carefully, the relevant studies were reviewed, and information was extracted and compiled from the literature. We focused on the relevant clinical trial data to summarize statistics on the clinical efficacy and AEs of multiple targeted agents for ATC patients. For promising targeted drugs that have not been studied clinically, we discussed them mainly from existing in vitro and in vivo trials.

## 3. Results

### 3.1. BRAF/MEK Inhibitors (BRAF/MEKi)

BRAF mutation is highly prevalent in malignant melanoma, non-Hodgkin lymphoma, and other diseases. It is also the most common somatic mutation in ATC, with V600E being the most prominent mutation site of the gene, particularly when ATC coexists with PTC, and more than 90% of patients may have BRAF V600E mutations [2].

#### 3.1.1. Dabrafenib plus Trametinib (DT)

Dabrafenib and trametinib are targeted drugs targeting BRAF and MEK1/2 belonging to the serine/threonine kinase family of the RAS pathway, respectively, which inhibit tumor cell proliferation by blocking the RAF-MEK-ERK signaling pathway. In 2017, it was demonstrated that combination therapy with dabrafenib and trametinib had significant clinical effects in ATC patients with BRAF V600E-mutant, with an overall response rate (ORR) of 69% and a one-year OS rate of 80%, whereas the same regimen in DTC patients with BRAF V600E-mutant had an ORR of only 33% [7,79]. The Food and Drug Administration (FDA) has therefore approved the combination of DT for the treatment of individuals with BRAF V600E-mutant ATC that is inoperable or metastatic. Recently, the phase II open-label study (ROAR) [8] published the latest clinical data (dabrafenib 150 mg twice daily and trametinib 2 mg once daily); at the time of data cut-off (about six years), the ORR reached 56%, with three patients responding completely, and the 12-month duration of response (DOR) was 50%. This long-term clinical trial confirmed that the combination of DT considerably lengthens the survival of ATC patients. In addition, a study [9] found that the combination of DT can be applied as the neoadjuvant therapy for ATC patients initially assessed as inoperable; patients benefited from conversion therapy followed by surgical resection and adjuvant therapy with this combination, with a local control rate of 100%, but the number of patients included in the study was too small. The viability of implementing this regimen as neoadjuvant therapy in the clinic is a topic worthy of future investigation.

The six-year clinical trial [7,8] showed that fever was the most common adverse event (AEs) with dabrafenib plus trametinib for ATC patients (47%), followed by anemia, fatigue, nausea, and anorexia, and eventually grade 3/4 AEs in 58% of patients, most commonly fatigue, anemia, and neutropenia. In severe cases, pneumonia, pleural effusion, acute kidney injury, and rhabdomyolysis may lead to permanent discontinuation. Similar to the results of the large COMBI-Aplus study [80], which demonstrated that the current clinical application of the DT combination for melanoma resulted in fever in 67.8% of patients, but the symptoms were generally mild, with an incidence of grade 3/4 fever of 3.8%, which could be effectively controlled by temporary discontinuation of the drug. Therefore, in clinical practice, clinicians must be aware of the occurrence of this AE. If the patient has a fever above 38 °C, immediately discontinue both drugs and apply the symptomatic treatment with non-steroidal anti-inflammatory drugs (NSAID) or low-dose hormone as appropriate; when patients’ temperature drops below 38 °C for 24 h, the drugs can be reintroduced; if there is still no response within 48 h after suspension of administration and symptomatic treatment, the infection index examination should be improved, and the use of antibiotics should be determined based on the results; repeated fever with poor control requires a decrease in the DT dose; when dose reduction and hormonal prophylaxis are ineffective, permanent discontinuation of the drug should be considered. Overall, these studies demonstrated that long-term therapy with DT was well tolerated and yielded favorable results.

#### 3.1.2. Vemurafenib

Vemurafenib belongs to a class of small-molecular weight BARF inhibitors that selectively inhibit oncogenic BRAF kinases. The FDA and the European Medicines Agency (EMA) have approved the targeted drug for treating patients with unresectable or advanced metastatic melanoma with BRAF V600E mutation. Hyman et al. [10] included seven ATC patients (5.7%) with BRAF V600E-mutant in a basket study of vemurafenib and showed an ORR of 29%, with two patients achieving complete response (CR) and partial response (PR), respectively. There are additional case studies [11,12] indicating that vemurafenib has high efficacy in certain ATC patients, with a prolonged response period of up to 61 weeks. In addition, Pilli et al. [13] evaluated the effect of vemurafenib in combination with TNF-related apoptosis-inducing ligand (TRAIL) to induce apoptosis in four ATC cell lines and showed that one of the ATC cells (8505C cells) was resistant to TRAIL and that this resistance could be decreased by vemurafenib. The ability of TRAIL to cause apoptosis in 8505C cells can be shown both in vitro and in vivo, resulting in the delayed growth of xenografts containing 8505C cells. This study suggests that vemurafenib is clinically effective in ATC patients with BRAF mutation and may also be an effective complementary therapy in cases of resistance to other drugs.

In this trial, including ATC patients mentioned above and another clinical trial of radioiodine-refractory (RAI-R) TC for BRAF V600E-mutant [10,81], the most common AEs of vemurafenib included rash, fatigue, arthralgia, alopecia, gastrointestinal reactions, hematologic toxicity, and serious AEs reported were cutaneous squamous cell carcinoma, keratoacanthoma, dyspnea, pneumonia, hypotension, and cerebrovascular accidents. In addition, non-cutaneous second malignancies have been recorded as AEs: distal squamous cell carcinoma of the trachea, squamous cell carcinoma of the head and neck, and gastric adenocarcinoma, which are considered to be probably connected to vemurafenib. However, there is no clinical investigation on the efficacy and safety of vemurafenib in only ATC patients, and the incidence of AEs caused by the medicine in ATC patients remains unknown. In practical use, however, care must be taken to avoid skin-related AEs and squamous metaplasia early and to reexamine blood routine tests regularly to detect bone marrow suppression in a timely manner so that patients can alleviate their suffering due to AEs.

### 3.2. NTRK Inhibitors (NTRKi)

TRK (tropomyosin receptor kinase) receptor family plays a crucial function in regulating cellular communication, encoded by the NTRK1, NTRK2, and NTRK3 genes, to produce the chimeric TRK fusion proteins (known as the oncogenic driver) that make up the activation pathway.

#### 3.2.1. Larotrectinib

Larotrectinib is an inhibitor of TRK (TRKA, TRKB, TRKC) that binds highly selectively to the protein product encoded by the NTRK fusion gene, blocking the activation and transmission of downstream signaling pathways to inhibit the growth and proliferation of tumor cells with this mutation. The FDA has approved the drug for treating adult and pediatric patients with solid tumors with NTRK fusions, and the application of this therapy is not restricted to a specific tumor type. In a study [14] that pooled data from three phase I/II clinical trials of larotrectinib, 7 patients (24%) with ATC finally had an ORR of 29% and a DOR of 50% at 12 months. Although the reported ORR rate was low, the treatment response exceeded that of previously reported with cytotoxic chemotherapy [2]. In one phase I trial of solid tumors with NTRK fusion in children [15], none of the two ATC patients receiving oral larotrectinib had progressed after seven months of treatment. Larotrectinib has minimal efficacy in ATC, and more clinical trial data are needed to support its actual effectiveness. 

AEs occurred in 90% of patients when taking larotrectinib [14]. Myalgia, fatigue, nausea, constipation, cough, dizziness, and elevated liver enzymes are common. Gastrointestinal and blood-related AEs can also occur frequently [14,15]. Patients should protect the gastric mucosa and pay attention to liver function and blood changes during application. However, the incidence of AEs of grade 3/4 due to larotrectinib is low, and the drug is relatively tolerated. Currently, there are no targeted results regarding the safety of larotrectinib for patients with ATC.

#### 3.2.2. Entrectinib

Entrectinib, similar to larotrectinib, is a broad-spectrum anticancer targeted agent that has been approved by the FDA in 2019 for adults and pediatric patients (older than 12 years old) with solid tumors with NTRK fusions and no known acquired resistance mutations. This indication includes TC [82]. A study [16] integrated and analyzed three phases I/II trials of entrectinib in patients with advanced or metastatic NTRK fusion-positive solid tumors, showing an ORR of 57%, a CR rate of 7%, and a PR rate of 50%, including 5 patients (9%) of TC who achieved an ORR of 20% and experienced tumor shrinkage, including two cases with a tumor reduction of more than 30% achieving PR. The specific therapeutic impact of entrectinib in ATC is unknown because the number of TC cases included in these studies is insufficient, and it is unclear whether they are all ATC patients. Nevertheless, it is suggested that patients with ATC undertake genetic testing and enroll entrectinib clinical trials if NTRK fusion is detected. In addition, in a study of patients with ROS1 fusion-positive advanced non-small cell lung cancer (NSCLC) with brain metastases, the final remission rate of intracranial lesions after treatment with entrectinib was 79.2% [83], suggesting that entrectinib is highly likely to cross the blood–brain barrier, which may also be beneficial for treating ATC with brain metastases.

According to recent studies [16,82], AEs can emerge as gastrointestinal reactions, hematologic toxicity and fatigue, dizziness, elevated blood creatinine levels, and taste disturbances in nearly all patients receiving entrectinib. More than half of the patients develop grade 3/4 AE, mainly pulmonary infections, dyspnea, pulmonary embolism, and other respiratory diseases; weight gain and cognitive impairment, congestive heart failure, central system reactions, and hepatotoxicity in severe cases. Monitor blood routine and liver and kidney function during application to avoid complications. In the event that a patient experiences chest tightness and shortness of breath, chest imaging tests and an electrocardiogram should be conducted immediately, and symptomatic treatment must be administered.

### 3.3. RET Inhibitors (RETi)

The proto-oncogene RET encodes a transmembrane tyrosine kinase, and the RET gene mutation is regarded as a significant contributor to the development of numerous cancers, including TC and NSCLC. Chromosomal rearrangements occur when the sequence encoding the intracellular kinase structural domain of RET is linked to the n-terminal protein dimeric structural domain sequence of another protein, when RET is usually allowed to be aberrantly expressed in transcriptionally silenced cells, resulting in RET fusions that activate multiple downstream signaling pathways, such as RAS, PI3K, and STAT, involved in cell growth and proliferation [84]. Certain rearrangements and mutations involve the frame fusion of RET with various chaperones; CCDC6 is the most common RET fusion chaperone in TC patients [85].

#### 3.3.1. Selpercatinib

Selpercatinib exhibits antitumor activity in cells containing harboring activation of RET proteins caused by gene fusions and mutations, including CCDC6-RET, KIF5B-RET, RET V804M, and RET M918T [85], designed to inhibit natural RET signaling as well as the expected mechanisms of acquired resistance. Selpercatinib was authorized by the FDA and EMA in 2020 for the treatment of RET-mutant medullary thyroid carcinoma (MTC) and RAI-R TC with RET-fused. In a phase I-II clinical research (LIBRETTO-001) [17], patients with RET-fused TC, comprising 143 treated and untreated MTC patients and 19 previously treated RET fusion-positive TC patients (including 2 ATC patients), were evaluated for the efficacy of selpercatinib. The results showed that these 19 patients had an ORR of 79% and a one-year progression-free survival (PFS) rate of 64%. One of the ATC patients had tumors that had metastasized to multiple sites throughout the body, including the brain and lungs, who had exhausted conventional treatment prior to selpercatinib administration and entered this clinical trial with a reduction in systemic lesions. At the time of publication of the follow-up report [18], this patient showed a sustained shrinkage in measurable lesions of 56.19% on computed tomography (CT) compared to baseline (according to RECISTv1.1), with clinical response to selpercatinib for more than 19 months.

The presence of xerostomia, hypertension, and digestive symptoms such as nausea, constipation, and bloating should be noted with selpercatinib treatment, and the most frequent grade 3/4 AEs include hypertension, elevated hepatic transaminase, hyponatremia, and diarrhea [17]. In a large clinical trial of selpercatinib in RET-fused NSCLC, hypersensitivity reactions were likewise frequent and severe, with the remaining AEs similar to those described previously [86]. Sepsis, cardiac arrest, pneumonia, respiratory failure, hemoptysis, postoperative bleeding, and heart failure were grade 5 AEs, but they were not considered by the investigators to be related to the drug [17,86]. Physicians should monitor the blood pressure and liver function, electrolytes of patients taking selpercatinib, and add medications to protect the gastric mucosa. AEs can usually be controlled by dose adjustment or discontinued if severe or requested by the patient (most discontinuation AEs are abnormal liver function and hypersensitivity reactions [86]). In the future, clinical trials of selpercatinib on the efficacy and safety of selpercatinib for ATC can be conducted, and additional analyses can be performed.

#### 3.3.2. Pralsetinib

Pralsetinib inhibits the phosphorylation of RET and its downstream molecules, hence suppressing the growth of cells containing variations of the RET gene. It has also been approved by the FDA and EMA for the treatment of adults and children (older than 12 years old) with advanced RET fusion-positive TC requiring systemic therapy and RAI-R TC (if radioactive iodine is applicable). Pralsetinib is the first drug approved for marketing in China as a RET inhibitor. In a multicenter, open, multicohort clinical trial (ARROW), pralsetinib was evaluated [19] in patients with metastatic RET fusion-positive TC and showed an ORR of 89%. However, the trial did not include ATC patients, and there is insufficient evidence to demonstrate the drug’s efficacy in ATC. Pralsetinib is promising for the treatment of ATC, as RET mutations can also be present in patients with ATC [20], and the drug is effective for treating other types of TC. Clinical trials on the efficacy and safety of pralsetinib in treating ATC patients with RET fusions or mutations could be initiated subsequently, and the drug could also be considered applied in the clinic in patients with advanced ATC who have failed to respond to conventional treatment modalities.

ARROW is comparable to a recent report on AEs in NSCLC patients, which demonstrated that pralsetinib induced hematologic toxicities, such as neutropenia, lymphopenia, and anemia, compared to selpercatinib, in addition to hypertension, elevated alanine transaminase (ALT), and aspartate transaminase (AST). Grade 3/4 AEs were commonly associated with hematologic toxicities and hypertension. However, severe treatment-related AEs are uncommon, with pneumonia being the most serious. Pneumonia is typically adequately treated with high-dose intravenous and/or oral corticosteroids. Just several patients discontinued therapy due to serious AEs, with 1% of patients dying from severe AEs considered treatment-related [19,87]. During the therapy period, the blood routine, blood pressure, and lung conditions should be monitored routinely, and timely symptomatic treatment should be administered. Overall, pralsetinib generated tolerable AEs and has enormous clinical application potential.

### 3.4. mTOR Inhibitors (mTORi)

mTOR is a serine/threonine protein kinase that is extremely conserved. mTOR belongs to the phosphatidylinositol-3-kinase-related kinase (PIKK) protein family because its C-terminus is highly homologous to the catalytic structural domains of PI3K and PI4K. The sustained over-activation of mTOR signaling will lead to increased levels of cellular metabolism, promoting continued cell growth and proliferation and even cell immortalization, which directly or indirectly induces cancer [88]. mTOR inhibitors block the mTOR signaling pathway, producing anti-inflammatory, anti-proliferative, autophagic, and apoptosis-inducing effects.

#### 3.4.1. Everolimus

Everolimus is a type of mTOR inhibitor, a hydroxyethyl derivative of sirolimus, which binds to an intracellular protein (FKBP-12) to form an inhibitory complex with the mTOR complex (mTORC1), thereby inhibiting the activity of mTOR kinase, and also reduces the expression of vascular endothelial growth factor (VEGF) then inhibits tumor angiogenesis [89,90]. A 2009 in vitro study indicated that everolimus could respond to a subset of ATC cell lines [21]. To assess the clinical efficacy of everolimus in patients with RAI-R TC, Lim et al. [22] initiated a multicenter clinical trial in patients with all histologic subtypes of locally advanced or metastatic TC, including 6 patients with ATC, who ultimately demonstrated a median PFS of 10 weeks, with one patient experiencing 21% tumor regression. A few years later, another phase II open-label trial [23] enrolled 50 patients with RAI-R TC taking oral everolimus 10 mg daily and showed an overall clinical benefit rate of 80%, with 1 of the 7 ATC patients (14%) maintaining disease stability for 26 months with no evidence of disease progression before death (due to congestive heart failure), and 4 patients progressed within three months of the study. In addition, one case achieved a near CR within 18 months and found that the anti-oncogene TSC2 in this patient’s pretreated tumor tissue contained a somatic nonsense mutation that reactivated the mTOR pathway, which may reveal a mechanism of drug resistance [91]. Harris et al. [24] performed a retrospective analysis of 5 ATC patients receiving everolimus as palliative therapy; the median survival of this cohort was 7.4 months, with one patient achieving PR within 27.9 months and two patients achieving stable disease (SD) at 3.7 and 5.9 months, respectively. Several clinical trials have demonstrated the clinical efficacy of treating ATC with PI3K/mTOR/Akt mutations, but further evidence from large trials on ATC is needed.

The majority of everolimus-treated ATC patients have AEs. Mucositis, rash, cough, anorexia, increased ALT/AST and anemia, thrombocytopenia, etc., are the most common grade 1/2 AEs. Common AEs of grade 3/4 include mucositis or stomatitis (7.8%), fatigue (4.4%), neutropenia (4.4%), infection (3.3%), hypercholesterolemia (2.2%), and pneumonia (1.1%), etc. Among these, pneumonia and hypercholesterolemia should be taken seriously since they caused two patients to discontinue treatment, respectively. Detailed in Table 1 [22,23]. In conclusion, the medicine is well tolerated. If everolimus is used clinically, the medical team should focus on the patient’s respiratory symptoms, monitor changes in blood lipids, and improve their nutrition as needed; oral and skin care should not be overlooked.

#### 3.4.2. Rapamycin

Rapamycin is a novel immunosuppressant with significant efficacy in autoimmune diseases and against organ transplantation rejection; as a targeted inhibitor of mTOR, the drug can also inhibit the proliferation and growth of tumor cells. In a previous study [25], ten cell lines of DTC and ATC were handled with the MEK1/2 inhibitor ADZ6244 (selumetinib) and rapamycin. The rate of growth inhibition was higher than 60% in six cell lines with rapamycin and greater than 60% in all ten cell lines with the combination of the two medicines (superior to either agent alone). Murugan [26] et al. further demonstrated that the prevalence of mTOR mutation in ATC cell lines was 12%, which was higher than that of DTC cell lines, and that ATC cell lines containing mTOR mutants were highly responsive to rapamycin at clinically relevant concentrations. These researches suggested that rapamycin shows a satisfactory anti-response in vitro to ATC. However, relevant in vivo studies are currently lacking, and this medication may be one of the future research targets for the treatment of ATC-targeted agents.

### 3.5. Anti-Angiogenesis TKI

TKIs function as an analog of adenosine triphosphate (ATP) or tyrosine to bind competitively to tyrosine kinase, which blocks tyrosine kinase to catalyze the phosphorylation of multiple substrate protein tyrosine residues and inhibit cellular signaling, sequentially inhibiting the proliferation and differentiation of tumor cells. ATC cells secrete vascular endothelial growth factor (VEGF), which leads to massive tumor neovascularization and increases the permeability of the vessel wall, hence facilitating cancer cell migration into the vasculature and metastasis [92,93]. Anti-angiogenesis TKIs can antagonize VEGF and control tumor angiogenesis.

#### 3.5.1. Sorafenib

Sorafenib, a novel multitargeted anticancer drug that targets RAF kinase, c-Kit and RET proto-oncogenes, VEGF receptor (VEGFR), and platelet-derived growth factor receptor (PDGFR), may exert its effects by arresting tumor cell cycles and by exerting anti-vascular effects [94,95]. Sorafenib inhibited the growth of ATC cells and improved their survival rate in preclinical animal models [27,28]. In contrast, the drug performed unsatisfactorily in a phase II trial involving patients with advanced ATC [29], with a median PFS of just 1.9 months, a one-year survival rate of 20%, and PR in two patients (10%). Ten ATC patients who took sorafenib in a Japanese clinical trial [30] had a median PFS of 2.8 months and a median OS of 5 months. The fact is that sorafenib has limited efficacy in treating ATC. Nevertheless, Chen [31] et al. found that sorafenib and metformin operated synergistically to reduce the proliferation of ATC cell lines and the growth of their derived ATC stem cells; the combination of the two medications had no effect on inhibitory function, even after reducing the dose of sorafenib. Another research [32] suggested that a histone deacetylase (HDAC) inhibitor (HDACi), N-Hydroxy-7-(2-Naphthylthio) Heptanomide (HNHA), which plus sorafenib combined with radiation therapy (RT) suppressed ATC cell growth more effectively than either medication combined with RT alone. Therefore, the combination of sorafenib with other treatments for ATC merits extensive research.

To summarize the two trials above on the specificity of sorafenib for ATC, the most prevalent AEs of sorafenib included skin reactions (83.3%), weight loss (56.7%), fatigue (50%), and anemia (36.7%). Grade 3/4 AEs were frequently for skin reactions, hypertension and hyponatremia, and elevated ALT/AST. One instance experienced gastrointestinal perforation, and one case was terminated permanently due to a rise in ALT. However, the overall toxicity was acceptable. The use of sorafenib should be supported by nutritional supplementation, warming and moisturizing of the skin, monitoring of blood pressure, and routine evaluation of liver function and electrolytes, as detailed in Table 2 [29,30].

#### 3.5.2. Lenvatinib

Similar to sorafenib, lenvatinib acts multi-targeted to VEGFR and PDGFR but not RAF and has a similar anticancer and anti-angiogenic effect, which has been shown to inhibit cell proliferation of ATC both in vitro and in vivo [33]. It is the only clinical agent approved for application in unresectable ATC patients in Japan [95]. Huang [34] et al. conducted a meta-analysis of the efficacy of lenvatinib for ATC patients, which included 10 relevant clinical trials over the past 25 years, and found that the combined PR, SD, and disease control rate (DCR) were 15%, 42%, and 63%, respectively. In one of these trials, a phase-II, international multicenter, open-label clinical trial [35] including 34 patients with ATC receiving lenvatinib (24 mg, once-daily), the final ORR was 0%, and the trial was terminated due to failure to achieve the desired goal (ORR < 15%). A recent Japanese study (HOPE) [36] evaluated the effectiveness of lenvatinib in unresectable ATC patients with an estimated one-year overall survival rate of 11.9%, an objective remission rate of 11.9%, and a clinical benefit rate of 33.3%, with a low number of responders to the drug but a more sustained response. Lenvatinib has the potential to be beneficial in the treatment of ATC; however, its clinical application is currently less efficient. Nonetheless, a number of studies have demonstrated that lenvatinib enhances the antineoplastic effects of adriamycin in ATC cells and xenograft models [37]. Lenvatinib can also synergistically enhances the antineoplastic effects when combined with anti-PD1/PD-L1 agents, vinorelbine, and paclitaxel [38,39,40], indicating that combining lenvatinib with other antineoplastic drugs may be more effective than single agents in controlling the progression of ATC patients.

The most frequent AEs were hypertension (55.6%), anorexia (53.5%), asthenia (37.2%), and proteinuria (34.8%); grade 3/4 AEs were unusual, and in a small percentage of patients, skin fistula and embolism, as well as life-threatening cases of tracheal perforation and mediastinal pneumothorax that may be caused by lenvatinib administration. In general, these AEs can be managed with dose reduction or discontinuation. Patients should also be advised to rest adequately, monitor their blood pressure and renal function, and be carefully managed and continuously monitored to avoid the occurrence of serious AEs while taking the targeted medicine, as detailed in Table 3 [41,42]. 

#### 3.5.3. Imatinib

Imatinib is commonly used in the clinic to treat chronic myelogenous leukemia (CML) and gastrointestinal stromal tumors (GIST); it is an inhibitor that targets BCR-ABL kinase, c-Kit receptor, and PDGFR. In in vitro experiments have shown that imatinib enhances the ability to suppress nuclear factor-B activation in ATC cells and effectively inhibits the proliferation of ATC cells [43,44]. Ha et al. [45] enrolled 11 patients of ATC taking oral imatinib (400 mg, twice daily) with PR of 25% and SD of 50% at 8 weeks but CR of 0%, an estimated 6-month PFS rate, and an ORR were 27% and 46%, respectively. Common AEs included abnormal liver function and myalgia (72.7%), electrolyte disturbance (63.6%), anemia and fatigue (54.5%), lymphopenia, cough, and dyspnea (45.5%), and grade 3/4 AEs frequently included lymphopenia (45.5%), edema (27.3%), anemia, nausea and vomiting, myalgia, and syncope (18.2%) [45]. Moreover, the combination of gefitinib and imatinib could strengthen the effect of resisting ATC [46]. Imatinib for ATC has not been investigated extensively in recent years; from the available literature, it can be concluded that the efficacy of this drug for ATC is limited, and its combination with other medications may become one of the future directions of research. The toxicity of imatinib in the treatment of individuals with ATC was comparable to that reported in previous studies; patients should pay special attention to cardiotoxicity, and those with hepatic insufficiency should continuously monitor liver tests and electrolytes [96].

#### 3.5.4. Sunitinib

Sunitinib is a low-molecular-weight TKI with multiple targets. Experiments have proved that sunitinib actively suppresses the growth of ATC cells in vitro and in vivo, which targets the MEK/ERK and SAPK/JNK signaling pathways [47]. However, several investigations have shown that sunitinib has little effect on the proliferation or differentiation of ATC cells [48]. A phase II multicenter clinical trial of sunitinib in patients with advanced DTC, metastatic ATC, and metastatic MTC [49] showed an objective response rate of 0% to ATC. During oral administration of this drug, patients reported asthenia (83.1%), diarrhea (60.6%), mucositis (64.8%), hand-foot syndrome (HRFS) (53.5%), and other skin-related AEs. Patients may experience life-threatening AEs of grade 5 involving diarrhea, bleeding, cardiac-related, respiratory-related, and vascular-related AEs when taking sunitinib [49]. Sunitinib monotherapy appears to be ineffective and poorly tolerated in the treatment of ATC; further trials are urgently required due to the paucity of previous research.

#### 3.5.5. Anlotinib

Anlotinib, the multi-kinase inhibitor that targets VEGFR/FGFR associated with angiogenesis, c-Kit kinases, and downstream signaling pathways mediated by PDGFR. The inhibitor prevents the formation of microvessels by acting on vascular endothelial cells through CXCL11-EGF-EGFR signaling even in hypoxia, interferes with ATC cell growth and proliferation, induces apoptosis, and inhibits migration [50,51]. Gui et al. [52] reported a 67-year-old patient with ATC who received anlotinib in combination with sintilimab when no other treatment alternatives were available and ultimately obtained maintained PR for 18.3 months; in this case, the patient experienced only a grade 1 rash and was well tolerated. These studies suggest that anlotinib is an emerging target drug against ATC and warrants more relevant in vitro and in vivo trials.

#### 3.5.6. Apatinib

Apatinib mainly acts on intracellular VEGFR-2 and inhibits receptor tyrosine kinases (RTK) such as c-kit and RET, mainly clinically applied in gastric cancer. Jin [53] et al. demonstrated that inhibiting the Akt/GSK3β/ANG signaling pathway by apatinib might play an essential role in preventing ATC angiogenesis. Feng et al. [54] further confirmed that apatinib induces autophagy and apoptosis by downregulating p-AKT and p-mTOR of the AKT/ mTOR signaling pathway in ATC cells. In a recent phase II, single-arm, open-label, Chinese clinical trial [55] evaluating the efficacy and safety of apatinib for ATC and PDTC (500 mg once daily), the final DCR was 88.2%, and the ORR was 41.2%. However, 64.7% of patients experienced at least one dose reduction due to grade 3 or higher AEs, and 23.5% discontinuation who unable to tolerate; the most common AEs were hypertension (88.2%) (grade 3 and above 47.1%), proteinuria (76.5%) (grade 3 and above 29.4%), HFSR (64.7%) (grade 3 and above 11.8%), leukopenia (47.1%) (grade 3 and above 5.9%), and thrombocytopenia, dyspepsia, mucositis, etc. [55]. In addition, the study also demonstrated that low-dose apatinib plus melittin works synergistically and can potentially reduce the incidence of AE [55]. Apatinib may have good efficacy in treating ATC, but the incidence of AEs is high. Patients should be applied active skin care, blood pressure monitoring, and regular review of urine and blood routine tests. Further relevant clinical evaluations can be conducted on apatinib, and methods to reduce the occurrence of AE can be explored.

#### 3.5.7. Vandetanib

Vandetanib is the first targeted medication specifically approved by the FDA for the treatment of MTC, acting not only on epidermal growth factor receptor (EGFR), VEGFR, and RET in cancer cells but also inhibiting other TK and serine/threonine kinases and possessing anti-angiogenesis properties. Ferrari et al. [56] proved that vandetanib induced apoptosis and significantly inhibited the proliferation, migration, and invasion ability of ATC cells; it inhibited EGFR, AKT, and ERK 1/2 phosphorylation and down-regulated cyclin D1 expression in ATC cells, and prevented tumor growth, microvessel density, and VEGF-A expression in 8305C tumor tissues of nude mice. Vandetanib is responsive to ATC cells both in vitro and in vivo, and we look forward to future clinical trials to further confirm its efficacy.

#### 3.5.8. Pazopanib

Pazopanib is a TKI that targets VEGFR-1, VEGFR-2, VEGFR-3, etc. In a phase II trial [57] for advanced ATC, pazopanib was found to be ineffective in humans, with a median time to progression (TTP) of just 66 days; however, it had strong anticancer effects on ATC in vitro. Patients taking pazopanib developed AEs such as fatigue (73%), anorexia (53%), diarrhea (47%), hypertension (40%), nausea (40%), and in severe cases causing grade 3–5 AEs such as hypertension (13%), pharyngodynia (13%), atrial fibrillation (6.7%), and thrombosis (6.7%) [57]. Isham et al. [58] found that pazopanib potentiated the cytotoxic effects of paclitaxel and that the combination enhanced anti-ATC cellular effects in vitro and in vivo. The experiment conducted by Teresa et al. [59] determined that the drug had a substantial synergistic effect when coupled with topotecan in ATC cell lines, resulting in a large decrease in the using dosage of both drugs and a reduction in their toxicity. Thus, pazopanib monotherapy may be ineffective in treating ATC, but combing it with other medications offers a novel strategy.

#### 3.5.9. Gefitinib

High expression of EGFR causes enhanced signaling downstream, promoting cell proliferation and inhibiting apoptosis; gefitinib is an EGFR-TKI that binds EGFR reversibly and plays a role in treating various cancers. Bradley et al. [60] demonstrated that EGFR was overexpressed in approximately 83% of ATC tissues; gefitinib inhibited ATC cell proliferation, induced apoptosis, and exhibited significant antineoplastic activity against ATC cells in nude mice. In phase II open-label trial [61] of gefitinib, which included 27 patients with locally advanced or metastatic RAI-R RTC, the median PFS was only 3.7 months, and the OS was 17.5 months for all patients, with only one achieving SD among the 5 ATC patients (18.5%) included. Gefitinib was not statistically beneficial in the treatment of ATC, but it was generally well tolerated, with just 11% of patients experiencing grade 3 AEs, with rash (52%) and diarrhea (41%) being the most prevalent, followed by nausea (19%) and anorexia (11%) [61]. However, the results must be confirmed by additional clinical trials.

### 3.6. CDK4/6 Inhibitors (CDK4/6i)

CDK4/6 inhibitors, commonly applied to treat breast cancer, are small molecule inhibitors that competitive binding ATP, which inhibit CDK4/6 from forming a complex with Cyclin D and block ATP binding, hence cutting off upstream growth signals and preventing the G1 to S phase transition of cells [97]. The deletion or mutation of the RB1 oncogene is a rare event in ATC, and the phosphorylation and inactivation of it are carried out by the cyclin D/CDK4 complexes, suggesting that therapies to inhibit the cyclin D/CDK4 complex may be effective in ATC. A study [62] showed that the CDK4/6 inhibitor, palbociclib, strongly inhibited ATC proliferation in a xenograft ATC model but was susceptible to developing resistance; whereas combined treatment with PI3K/mTOR and CDK4/6 inhibitors synergistically reduced cell proliferation and was highly influential in inhibiting tumor growth even in cell lines that did not carry PI3K activating mutations or apply a low dose of the two drugs. The combination of PI3K/mTOR and CDK4/6 inhibitors is a promising new research direction for the treatment of ATC.

### 3.7. Other Targeted Agents

Combretastatin A4 disodium phosphate (CA4P) is an antineoplastic agent that targets critical angiogenesis molecules. Joshua et al. [63] compared the cytotoxic effects of CA4P and paclitaxel on eight human ATC cell lines; the results showed that CA4P exhibited significant cytotoxicity and lasted longer than paclitaxel in two cell lines; CA4P also reduced the rate of tumor growth and decreased tumor volume in nude mice. In the phase I clinical trial [64], a patient with ATC treated with CA4P attained durable CR (30 months). Later, Mooney et al. [65] included 26 ATC patients using CA4P (45 mg/m^2^) and observed the efficacy; the final overall median survival was 4.7 months, including 7 patients with PFS of 12.3 months and well tolerated, the main toxicities were cardiovascular related, such as hypertension, arrhythmia, QTc prolongation, and other AEs including fatigue, gastrointestinal reactions, and leukopenia; attention should be paid to electrocardiogram and blood pressure when using CA4P. In addition, Yeung et al. [66] discovered that the combination of CA4P and paclitaxel was more effective against ATC in nude-mouse xenograft models. Coincidentally, in the FACT clinical trial [67], which compared carboplatin/paclitaxel combing with and without CA4P in patients with ATC, the median survival in the CA4P group was 8.2 months, compared to 4 months in the control group; the 1-year survival rate was also higher in the CA4P group than in the control group. These studies indicate that CA4P has potential and that the drug may be effective in the treatment of ATC when combined with or without other agents with controllable AEs. CA4P is an innovative and promising treatment for ATC.

Previous research has demonstrated that valproic acid (VPA), one of the HDAC inhibitors, enhances the cytotoxic effect of adriamycin and paclitaxel in vitro to inhibit the proliferation of ATC cells [68,69]. However, the combination of paclitaxel with VPA did not prolong the overall survival in ATC patients in a randomized phase II/III clinical trial [70]. Despite the disappointing results of this trial, it appears that HDAC inhibitors may still be investigated as a novel target drug for ATC. Kim et al. [71] revealed that HDAC inhibitors and the heat shock protein 90 (hsp90) inhibitor SNX5422 synergistically improved their cytotoxic effects, inhibited PI3K/Akt/mTOR signaling and overexpression of DNA damage-related proteins in ATC cells. In addition, other HDAC inhibitors, such as suberoyl anilide hydroxamic acid (SAHA) and HNHA, also induce anti-ATC cellular effects, providing new hope for treating ATC [32,72].

Wächter et al. [73] applied selumetinib, a MEK inhibitor, to four different types of TC cell lines, revealing that all cell lines had their viability reduced by more than half, and ATC cell lines with the BRAF V600E mutation were the most sensitive, with a reduction of more than 80% in viability.

An experiment [74] demonstrated that cetuximab (IgG1 monoclonal antibody against EGFR) and bevacizumab (IgG1 monoclonal antibody against VEGF) alone or in combination inhibited tumor cell growth and angiogenesis in in vivo models of ATC. In addition, Kim et al. [75] showed that the combination of cetuximab and irinotecan suppressed the growth of xenograft ATC tumors and significantly reduced the occurrence of tumor metastases.

Peroxisome proliferator-activated receptor γ (PPARγ) agonists are also potential agents that promote cell cycle arrest, induce apoptosis, and are anti-angiogenic. RS5444, a PPARγ agonist, has been demonstrated to inhibit ATC cell proliferation in vitro and in vivo without inducing apoptosis, for which RhoB may be a key target [76]. Troglitazone and ciglitazone also belong to this class of medications, which increase the apoptosis rate of ATC cells and prevent the growth and spread of tumor cells [77]. However, no clinical trials related to the therapeutic regimen of ATC with these drugs have now been reported, either alone or in combination. Selumetinib, cetuximab, or PPARγ could be the following research targets for ATC treatment approaches based on a synthesis of the aforementioned.

According to previous studies, BRAF mutations occur in up to 20–50% of ATC, and RAS mutations occur in 50% [78]; however, only about less than 1–2% of ATC patients may have NTRK or RET gene fusion mutations [14]. Moreover, one study [98] found that increased gene copy number occurred in ATC at a frequency of about 46.3% for EGFR, 23.9% for PDGFR, and 45.5% for VEGFR. The results of primary endpoints in several major clinical trials and the primary target points of targeted agents for ATC are summarized in Table 4.

## 4. Discussion

In conclusion, the combination of DT has been identified as the most effective targeted drug for the current treatment of ATC patients with BRAF V600E mutated, with tolerable adverse effects, fever being the most common. NTRKi and vemurafenib, the other BRAF/MEKi, show limited clinical response in ATC. The efficacy of lenvatinib may not be as satisfactory as anticipated; other angiogenesis TKIs may be ineffective when taken alone; when coupled with chemotherapeutic or anti-PD1/PD-L1 agents, the clinical benefit rate is significantly increased. Based on the findings of basic research, RETi, mTORi, CDK4/6i, and CA4P are targeted agents with substantial therapeutic potential, despite the lack of clinical trials currently. All stated agents have a relatively high incidence of adverse effects, notably skin toxicity, gastrointestinal reactions, suppression of bone marrow, decreased liver and renal function, and hypertension, but they are controllable. When administering the medications, monitoring and management should be emphasized, and symptomatic treatment should be administered in the occurrence of AEs, which can be managed by dose reduction or withdrawal. Targeted therapy has tremendous potential for treating patients with ATC. However, more relevant cases and clinical trials are required to evaluate the specific and completely accurate efficacy of various agents for ATC.

On the other hand, drug resistance has become a major concern following the inefficacy or benefit of targeted drugs, even in the most practical combination of DT, which may be associated with the emergence of RAS mutations [99]; vascular cell adhesion molecule-1 (VCAM-1) and S100A4 protein have also been proven to be responsible for the resistance of vemurafenib [100,101]. However, the application of L-neutral amino acid transporter 1 (LAT1) inhibitors and the upregulation of the oncogene MiR-99a has been found to be further resistant to ATC cells by inhibiting the mTOR signaling pathway [102,103]. These findings imply that the discovery of potential drug resistance mechanisms and methods for resolving or reducing resistance is also necessary to enhance the efficacy of existing targeted medications and provide new directions for the development of new-generation targeted drugs. Combining targeted drugs with specific other medications may also improve effectiveness and decrease drug resistance [62], and further in vitro and in vivo studies and clinical trials are expected.

Due to the aggressiveness of ATC, all ATC patients were clinically classed as stage IV. When patients are diagnosed with the disease, they are generally inoperable, and traditional anticancer treatments such as chemotherapy and radiotherapy have minimal efficacy; thus, it is advantageous for them to participate in clinical trials of various targeted drugs actively. However, because of the rapid rate of invasion and metastasis, when diagnosing ATC patients for the first time, physicians must promptly adopt a therapeutic regimen and persuade patients to accept traditional systemic treatments as soon as possible to avoid a delay in patients’ condition. Simultaneously, surgical specimens or fine needle aspiration (FNA) tissue samples of the thyroid should be sent for genetic testing; based on the results, the right targeted medicine can be chosen for the treatment of ATC patients.

## 5. Conclusions

ATC is an aggressive and rapidly lethal tumor, and the efficacy of conventional treatments is very limited; however, surgery, systemic chemotherapy, and local radiotherapy should be used if possible. In the meantime, targeted therapy should be considered. As discussed in this review, the development of targeted drugs for the treatment of ATC is rapid, and most of their adverse effects are controllable. Compared to other targeted drugs, the combination of dabrafenib and trametinib is the most effective to date. Furthermore, combination therapy, especially which includes TKIs, can be regarded as one of the most important future research directions. It is worth noting that drug resistance is a critical and unavoidable problem for researchers and clinicians; studies have demonstrated that anticancer-targeted drugs in combination with other specific drugs may be able to reduce drug resistance of the former. Overall, targeted agent represents a promising approach to treating ATC. However, more basic experiments and clinical trials are needed to further demonstrate the efficacy and adverse effects of various targeted drugs or to study drug resistance issues for this fatal cancer. 

## Figures and Tables

**Figure 1 cancers-15-00179-f001:**
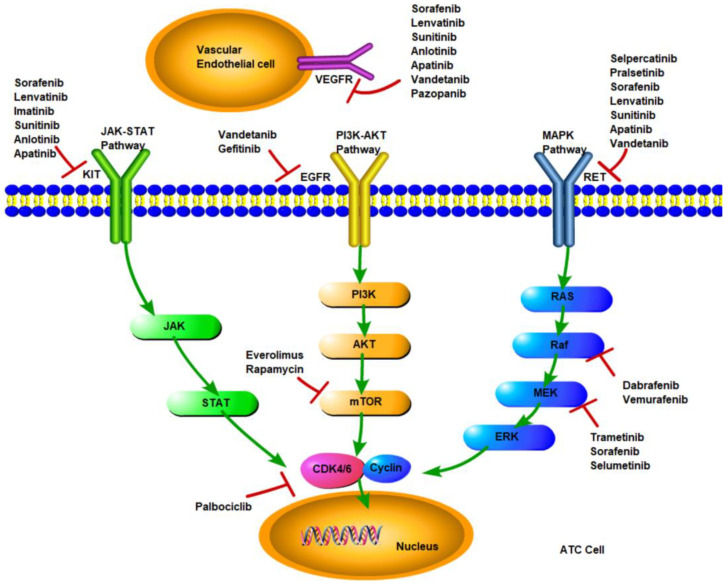
Signaling pathways and molecular targeted inhibitors in anaplastic thyroid cancer.

**Figure 2 cancers-15-00179-f002:**
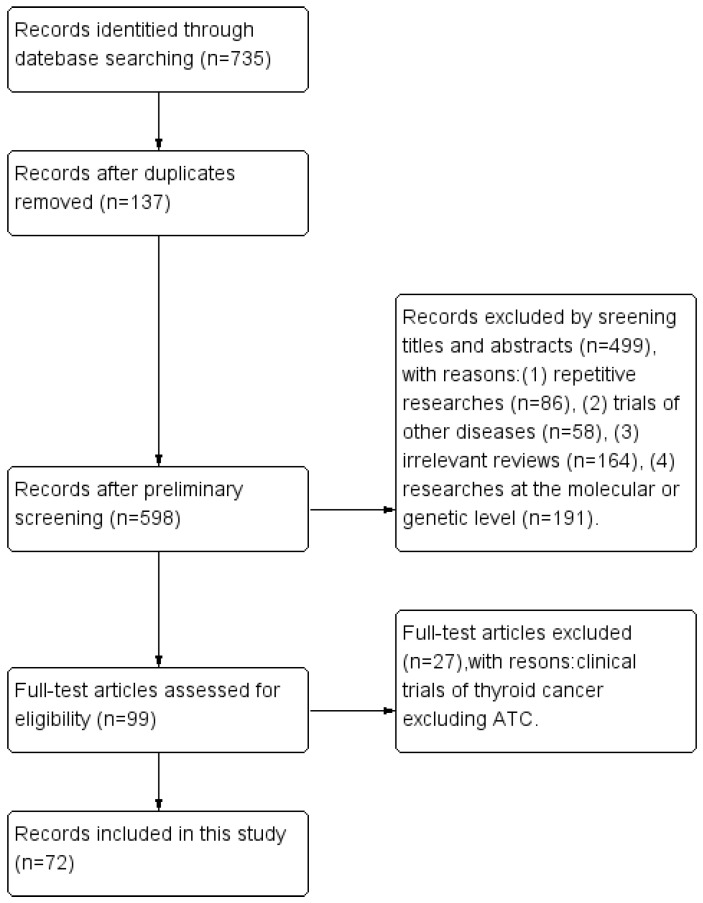
Flow diagram.

**Table 1 cancers-15-00179-t001:** Grade 3 or grade 4 adverse effects of Everolimus for ATC patients *.

Adverse Events, *n* (%) *	Grade 3	Grade 4
Mucositis or stomatitis	7 (7.8)	0
Fatigue	4 (4.4)	0
Neutropenia	4 (4.4)	0
Diarrhea	4 (4.4)	0
Infection	3 (3.3)	0
Weight loss	3 (3.3)	0
Anorexia	2 (2.2)	0
Hypertriglyceridemia	2 (2.2)	0
ALT/AST increase	2 (2.2)	0
Thrombocytopenia	2 (2.2)	0
Pneumonia	1 (1.1)	1 (1.1)
Hypercholesterolemia	0	1 (1.1)

* Specific data for Grade 1/2 AEs were unavailable for the two included clinical trials; the outcomes did not completely target ATC patients but were only contained; *n*, number; References [22,23].

**Table 2 cancers-15-00179-t002:** Adverse effects of Sorafenib for ATC patients *.

Adverse Events, *n* (%) *	Any Grade	Grade ≥ 3
Cutaneous reaction	25 (83.3)	3 (10)
Weight loss	17 (56.7)	1 (3.3)
Fatigue	15 (50)	1 (3.3)
Anemia	11 (36.7)	0
Diarrhea	10 (33.3)	0
ALT/AST increase	10 (33.3)	3 (10)
Hypertension	9 (30)	2 (6.7)
Stomatitis	9 (30)	0
Hyponatremia	9 (30)	2 (6.7)
Hypocalcemia	8 (40)	1 (3.3)

* The essential data of patient numbers for multiple AEs included in the two clinical studies are different; *n*, number; References [29,30].

**Table 3 cancers-15-00179-t003:** Adverse events of Lenvatinib for ATC patients.

Adverse Events, *n* (%) *	Any Grade	Grade ≥ 3
Hypertension	75 (55.6)	17 (12.6)
Anorexia	46 (53.5)	5 (5.8)
Asthenia	45 (37.2)	0
Proteinuria	39 (34.8)	3 (2.7)
Hypothyroidism	19 (34.0)	0
Fatigue	45 (33.3)	1 (0.7)
Stomatitis	19 (26.0)	1 (0.7)
Thrombocytopenia	12 (21.4)	0
Vomit	11 (10.2)	1 (0.7)

* *n*, number; References [41,42].

**Table 4 cancers-15-00179-t004:** Primary target points and summary of clinical trials of targeted agents *.

Targeted Agent		Primary Target Point	Reference	ATC (n/Total)	The Primary End Point	Result
BRAF/MEKi	DT	BRAF/MEK	[7]	16/16	ORR	69%
			[8]	36/36	ORR	56%
	Vemurafenib	BRAF	[10]	7/122	ORR	29%
	Selumetinib	MEK	NA			
NTRKi	Larotrectinib	TRK	[14]	7/29	ORR	29%
	Entrectinib	TRK	[16]	NA	ORR	20%
RETi	Selpercatinib	RET	[17]	2/19	ORR	NA
	Pralsetinib	RET	NA			
mTORi	Everolimus	mTOR	[22]	6/40	DCR	NA
			[23]	7/50	Median PFS	2.2M
			[24]	5/5	DCR	60%
	Rapamycin	mTOR	NA			
TKI	Sorafenib	VEGFR, PDGFR,Kit,RAF,RET	[29]	20/20	RECIST Response	PR 10%; SD 25%
			[30]	10/10	Median PFS	2.8M
	Lenvatinib	VEGFR,PDGFR, Kit,RET	[35]	17/17	ORR	0%
			[36]	42/42	one-year OS rate	11.9%
	Imatinib	BCR-ABL,Kit,PDGFR	[44]	11/11	DCR	75% (PR25%; SD50%)
	Sunitinib	PDGFR,VEGFR,Kit, FLT,RET	[49]	4/71	ORR	0%
	Anlotinib	EGFR,FGFR, PDGFR,Kit	NA			
	Apatinib	VEGFR,Kit,RET	[55]	17/17	DCR	88.2% (PR41.2%; SD47%)
	Vandetanib	EGFR,VEGFR,RET	NA			
	Pazopanib	VEGFR	[57]	15/15	RECIST Response	NA (poor)
	Gefitinib	EGFR	[61]	5/27	ORR	0%
CDK4/6i	Palbociclib	CDK4/6	NA			
CA4P		VEGFR	[65]	26/26	Median OS	4.7M
HDACi	VPA, SAHA,HNHA	HDAC	NA			
Cetuximab		EGFR	NA			
Bevacizumab		VEGFR	NA			
RS5444		PPARγ	NA			

* NA not available; M months; DCR PR + SD ≥ 8 weeks.

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
