# Peer review of "Targeted Therapy for Anaplastic Thyroid Carcinoma: Advances and Management"

_cancers, 2022, doi:10.3390/cancers15010179_

Round 1
Reviewer 1 Report
This review is of potential interest for clinicians exposed to anaplastic thyroid carcinoma (ATC) patients, regardless of subspecialty. It seems to cover most of the recent developments in targeted therapies for ATC. There should, however, be some additions made to the manuscript, thereby making it scientifically more robust.
1) Please describe the methodology for retrieving data for the review. Search terms? Mesh words? Which databases were used? How many articles were identified? Which articles were not included, and for what reasons? How were the final articles used for the various targets? Please create a flow chart for the process of identifying adequate articles, with numbers of articles included and excluded. At least a short description of Methods used should be mandatory.
2) There are plenty of targeted therapies that rely on certain genetic aberrations. It would be valuable with an added table describing all molecular targets identified so far, not only those that have been used in clinical trials (Table 4). Included in such a table, showing the proportion of ATC:s displaying such aberrations would be helpful, i.e. what percentage of ATC:s display RET aberrations, NTRK, etc etc. This can then be extrapolated to an estimated number (or fraction) of ATC patients who should potentially benefit from a specific targeted drug.
3) All tables in a review should be possible to read and interpret separate from the text. Please expand the tables’ headline, describing what information is given. Also, explain the shortenings in each table. In Table 1-3 it should be clarified from which references data were retrieved.
Author Response
Dear Reviewer,
Thank you for your comments concerning our manuscript entitled “Targeted Therapy for Anaplastic Thyroid Carcinoma: Advances and Management” (cancers-2034855). Those comments are all valuable and very helpful for revising and improving our paper. We have studied comments carefully and have made correction which we hope meet with approval. Revised portion are marked up with the “Track Changes” function in the paper. The main corrections in the paper and responds to the reviewers’ comments are as following:
Responds to the reviewer’s comments:
- Response to comment: Please describe the methodology for retrieving data for the review. Search terms? Mesh words? Which databases were used? How many articles were identified? Which articles were not included, and for what reasons? How were the final articles used for the various targets? Please create a flow chart for the process of identifying adequate articles, with numbers of articles included and excluded. At least a short description of Methods used should be mandatory.
Response: As suggested by the reviewer, we have created a flow chart that specifically describes the inclusion and exclusion process of articles, and described the search terms, databases and the inclusion and exclusion criteria, etc. in the part of Materials and Methods.
- Response to comment: There are plenty of targeted therapies that rely on certain genetic aberrations. It would be valuable with an added table describing all molecular targets identified so far, not only those that have been used in clinical trials (Table 4). Included in such a table, showing the proportion of ATC:s displaying such aberrations would be helpful, i.e. what percentage of ATC:s display RET aberrations, NTRK, etc etc. This can then be extrapolated to an estimated number (or fraction) of ATC patients who should potentially benefit from a specific targeted drug.
Response: Thanks for your suggestions very much, according them, we have added a column to Table 4 specifically listing the primary targets of various targeted drugs, whether or not they have been clinically tested. As for the percentage of all ATC patients accounted for by, for example, RET or NTRK fusion mutations, we have searched the relevant literature comprehensively, but we have chosen to include the relevant data within the text (Page 14, 593-599) because there are no recognized and accurate data to date. We hope for your approval.
- Response to comment: All tables in a review should be possible to read and interpret separate from the text. Please expand the tables’ headline, describing what information is given. Also, explain the shortenings in each table. In Table 1-3 it should be clarified from which references data were retrieved.
Response: Thanks for your suggestion, we have expanded the tables’ headline and explain the abbreviations in each table. We have indicated the references in the footnotes of the Table 1-3 and in the citations above.
We appreciate for your warm work earnestly, and hope that the correction will meet with approval.
Once again, thank you very much for your comments and suggestions. We look forward to hearing from you regarding to any further questions and comments that you may have.
Yours sincerely,
Jiaqian Yuan
17, Dec, 2022
Yuanjiaqian111@126.com

Reviewer 2 Report
The authors provide a comprehensive review of current treatment modalities of anaplastic thyroid cancer. Considering the aggressive nature and the poor prognosis of these patients, this study is a good addition to the current literature. The text is well written and organized, easy to follow and read.
Author Response
Dear Reviewer,
Thank you for your comments concerning our manuscript entitled “Targeted Therapy for Anaplastic Thyroid Carcinoma: Advances and Management” (cancers-2034855). Revised portion are marked up with the “Track Changes” function in the paper.
We appreciate for your warm work earnestly, and hope that the correction will meet with approval.
Once again, thank you very much for your comments. We look forward to hearing from you regarding to any further questions and comments that you may have.
Yours sincerely,
Jiaqian Yuan
17, Dec, 2022
Yuanjiaqian111@126.com

Reviewer 3 Report
The present review manuscript titled “Targeted Therapy for Anaplastic Thyroid Carcinoma: Advances and Management” by Yuan and Guo is novel and very well written. In this manuscript, the authors discussed different targeted therapy for anaplastic thyroid carcinoma. Overall, the quality of the manuscript is excellent. However, I have some suggestions for authors. My comments are as follows.
Comment 1. It would be better to discuss molecular targets for anaplastic thyroid carcinoma in a separate section.
Comment 2. The authors should diagrammatically represent different signaling pathways involved in anaplastic thyroid carcinoma.
Comment 3. The conclusion section is very poor. The authors should revise this section to improve the overall quality of the manuscript.
Author Response
Dear Reviewer,
Thank you for your comments concerning our manuscript, we have tried our best to revise the manuscript. Please see the attachment.

Round 2
Reviewer 1 Report
The authors have responded satisfactory to the issues raised.
Reviewer 3 Report
The authors addressed my comments very critically. I don't have further comments